# Style-Quizzes for Content-Based Fashion Recommendation in Extreme Cold Start Scenarios

Anonymous Full Paper
Submission 26

## Abstract

This article presents Style-Quiz, a novel method for circumventing the user-based cold start problem in the context of content-based recommender systems. We construct a content-based recommender system for a given environment and generate a quiz built upon its underlying embeddings. During the course of the quiz, the embedding space of the recommender system is segmented via unsupervised hierarchical clustering. The user is presented with a series of images representative of each cluster and tasked with choosing one of them. The chosen cluster is then segmented in the same way as its parent cluster. This process is repeated until the user has honed in on a point in the embedding space that adequately represents that user's tastes.

As a user interested in renting or purchasing fashion items is likely to be interested in several different kinds of fashion articles, we also introduce Style-Vectors. A representation of our items, built on deep-learning encoders and triplet loss, that is indicative of their underlying style, not just physical attributes.

Our results indicate that Style-Quiz significantly improves early personalized recommendation as compared to recommending globally popular items.

To improve reproducibility, we publish both the code and dataset used for the project.

## 1   Introduction

When developing Recommender Systems (RSs) against offline performance metrics such as Mean Average Precision at N (MAP@N) or Root Mean Squared Error (RMSE), like is common in most RS competitions[1][2], it's easy to overlook the challenges associated with live recommendations. Among the most prominent of these is the cold start problem for new users. Existing RSs tend to base their recommendations on previous user activity, including the top-performing submissions for the aforementioned competitions[3–5][6]. However, this approach to providing recommendations ignores the most crucial part of the customer base's user experience, namely the onboarding of new users. RSs that are reliant on previous user activity will, of course, not be able to provide meaningful recommendations to a customer without any previous user activity. This is referred to as the cold start problem. When absolutely no other information about a user is known beforehand, this is referred to as the extreme cold start problem.

The use of style quizzes to onboard new users is reasonably common within fashion retail. Two examples of companies using this method are Nordstrom[7] and Stitch Fix[8]. Though remarkably little has been written about this subject from an academic perspective. To our knowledge, this paper is the first article to discuss the generation of such a quiz in the context of RSs.

The work discussed in this article builds off of a dataset and existing work involving recommendations for the domain of fashion rental. The extreme cold start problem is particularly relevant to fashion rental due to its incompatibility with one of the most common methods for circumventing the cold start problem, namely to recommend the globally most rented items to new users[9]. Though recommending globally popular items can be a reasonably effective method for collecting rental history data on new users, this approach concentrates user attention on a certain set of items. i.e. as an item cannot be rented for a given period to two separate users, this attention will naturally cause undesirable competition between them. In which one of them is unable to rent the item he or she is most interested in for the period. This may lead this customer to feel that a company's inventory is more limited than what is actually the case.

This paper presents an RS method capable of performing in the context of both cold-start items and users. It does so by integrating a content-based RS with a procedurally generated quiz. The structure of content-based RSs ensures they are resilient against the item-based cold start problem, and Style-Quiz compensates for content-based RSs vulnerability to the user-based cold start problem. We also introduce Style-Vectors, a representation of items built upon a neural-item encoder and triplet loss. These representations are presented directly to the user during the onboarding phase as a style quiz in the form of their corresponding product

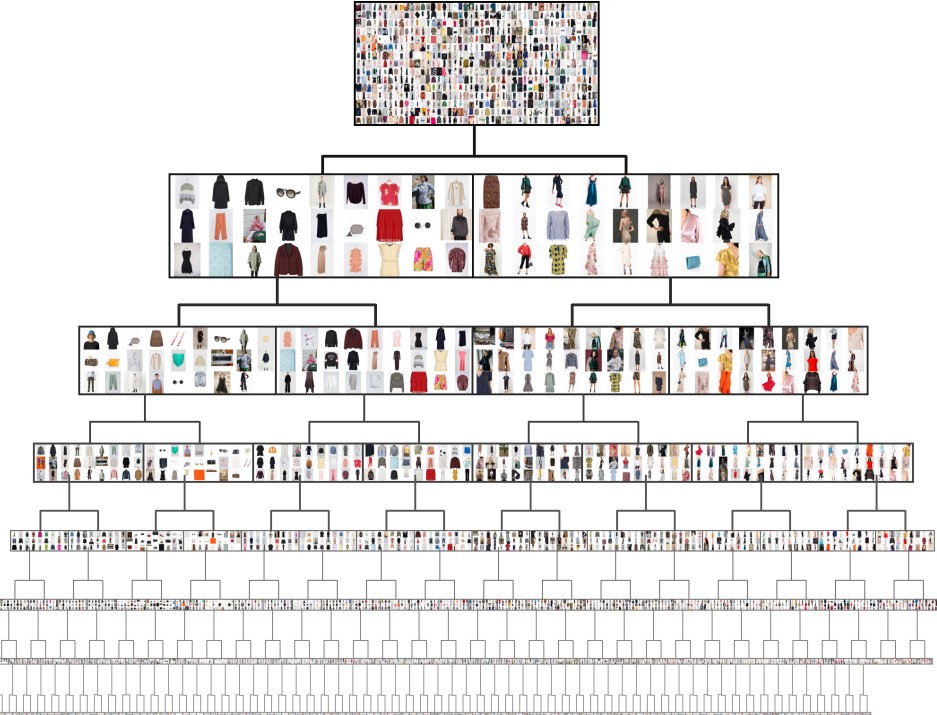

**Figure 1.** Visual representation of the decisions presented to the user. Each cluster is represented by a collage of the items most representative of that cluster.

images. The quiz dynamically adapts its questions to the user's answers, thereby gradually honing in on an initial estimation of the user's preferences. In this process, the user is exposed to a large section of the company's catalog, effectively making this approach to onboarding a more extreme kind of active learning[10].

## 2  Related Work

Recommendations of fashion, more specifically fashion retail, is a largely well-mapped domain. Deldjoo et al. provide a thorough and recent review of the state of the art, though with only a few mentions of the different methods' performance in cold start scenarios[11]. Elahi et al. provide a brief overview of the state of the art of the cold start problem within fashion RSs as of early 2021[12].

One potential approach for circumventing the user-based cold start problem is incorporating personality tests into the onboarding process. The fact that individuals with similar Big 5 personality profiles tend to be interested in the same kind of items is well documented within the literature[13][14]. Some implementations of personality-based RSs apply this representation of a user's personality to detect users with similar preferences. Another approach is to rapidly integrate the implicit feedback of new users based on clicks, wants, and purchases from the initial browsing sessions to rapidly generate early personalized recommendations[15]. This method is generally used in conjunction with initially recommending globally popular items, which is less than ideal, particularly in the context of fashion rental. In non-extreme cold start scenarios in which we have some knowledge about the users beforehand, the application of social media information may be used to onboard new users. A review of 10 such methods has been written by L. A. Gonzalez Camacho and S. N. Alves-Souza[16].

Several fashion retail sites deploy style quizzes in their onboarding process[7, 8], but to our knowledge, nothing has been written about these methods from an academic perspective.

## 3  Method

This project is built off of the Vibrent Clothes Rental dataset, a dataset detailing all rentals made from a small fashion rental company based in Norway. The dataset consists of 9791 outfits, 50293 images (with pre-computed embeddings), and 2249 user rental histories. Each of these outfits has a set of images and a set of tags associated with them. The tags denote some categorical attributes associated with the clothing piece. For example, the category "Color" could have the value "Blue," the category "Occasion" could have the value

"Everyday," and the category "Material" could have the value "Wool".

The dataset used for this work is the Vibrent Clothes Rental dataset[9] which is publicly available at https://www.kaggle.com/datasets/kaborg15/vibrent-clothes-rental-dataset.

The code for this project has been published via Github at https://anonymous.4open.science/r/Style_Quiz-B7B3/

## 3.1 Building Style-Vectors

To represent each outfit, we one-hot-encode its tag embedding and concatenate them to their leading image embedding. If we chose to encode all outfits based solely on this image and tag representations, the resulting clusters would be heavily influenced by the outfit's category, e.g., whether it's a dress, a purse, or trousers. As demonstrated by Borgersen et al.[17], even zero-shot image embeddings tend to cluster outfits of a similar category closely together and vice versa. On the other hand, the customers in our dataset tend to prefer diversity in the kinds of items that they rent. The mean Simpson's Diversity Index score[18] across all customers is 0.327, indicating that the probability that two outfits rented by the same customer belong to the same category is around 32%. Ideally, our internal representations should represent an outfit's style. So, outfits of different categories that could be worn on similar occasions or in combination with each other. For example, an elegant dress and a matching purse should be considered similar in our embedding space. In contrast to a heavy skiing jacket and a set of high heels, which should be considered far apart.

To compensate for this biasing towards similarity between outfit categories, we introduce a criterion that incentivizes the embeddings to maintain a structure representing an outfit's user-perceived style rather than just its physical traits. This is done via the application of triplet loss[19] in a neural-network-based item-encoder. The positive examples are outfits that have previously been rented by the same user, and the negative examples are retrieved randomly from the remaining list of outfits. A collage of the images representing each item is displayed along with their relative position to each other in Figure 3.

## 3.2 Procedurally Generating the Quiz Questions

The basic loop of the procedurally generated quiz from the perspective of the user is presented in Algorithm 1. We segment our embedding space and gradually partition it into smaller and smaller segments until it eventually converges upon a cluster small enough to be considered representative of a user's preferences. This implementation counts the quiz as having converged once a cluster contains fewer than 30 samples, though this number is arbitrary. The segmentation of the embedding space is performed via the hierarchical single linkage clustering. Essentially, this method forms a tree structure of clusters by gradually joining two adjacent clusters into a parent cluster based on the ward metric[20]. The distances between each cluster are given by the equation

$$d(u,v) = \sqrt{\frac{|v|+|s|}{T}d(v,s)^2 + \frac{|v|+|t|}{T}d(v,t)^2 - \frac{|v|}{T}d(s,t)^2}$$

Where $u$ refers to a cluster joined by $s$ and $t$, $v$ is an unused cluster and $T$ is the combined cardinality of all three of these clusters.

A visual representation of how the quiz is presented is displayed in Figure 1. To adequately represent the diversity of different items present in any segment, KMedoids[21] clustering is applied to find the k most representative items in the cluster. KMedoids functions similarly to KMeans, but rather than returning an arbitrary point in the embedding space as a representation of each cluster, KMedoids centers clusters on one of the exisiting items in the dataset. We use the images that represents these items as a representation of the entire cluster to the user. In this case, k is the number of images used to present each cluster.

# 4 Results & Discussion

The motivation for presenting the user with a set of different questions to parse through is to better represent the user within our content-based embedding space. Figure 2 displays all points in our embedding space condensed into two dimensions via T-distributed Stochastic Neighbor Embedding (T-SNE)[22], the points at which our quiz could converge are displayed as Xs. Based on visual inspection, we can see that the convergence points are evenly distributed across the embedding space.

Within a content-based RS, the preferences of a user can be represented as the mean embedding of their outfit rental history. As one of the most common approaches for circumventing the cold start problem is to recommend the globally most popular items in the environment, we evaluate our method by comparing our representations to the default most popular items. Across all of our 2249 users, the mean distance to the center of the most popular cluster is 255% longer than the closest quiz convergence point. To put this percentage into perspective, the mean number of items that exist within the radius between a user's point and the

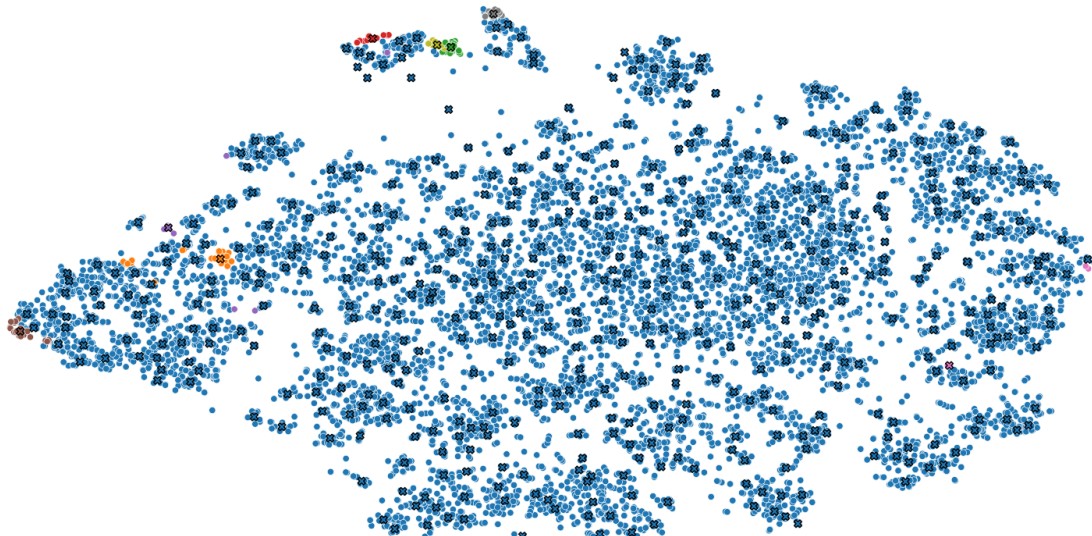

**Figure 2.** TSNE diagram of our data with convergence points. The potential convergence points of Style-Quiz are marked with an X. Nine of the clusters formed around the convergence points are marked with colors other than blue.

---

**Algorithm 1** Style-QuizLoop

---

1: **procedure** PARTITIONEMBEDDINGS($E$, $converged = 30$, $num\_images = 27$)
2:     **while** $|E| \geq converged$ **do**
3:        $\{E_1, E_2\} \leftarrow$ split_cluster($E, k = 2$)
4:        $R_1 \leftarrow$ KMedoids($E_1, K = num\_images$) $\triangleright$ Choose which points to use to represent the cluster
5:        $R_2 \leftarrow$ KMedoids($E_2, K = num\_images$)
6:        $E \leftarrow$ UserChoosePartition($E_1, E_2, R_1, R_2$)
7:     **end while**
8:     **return** $E$
9: **end procedure**

---

most popular item point is 113.2, and the mean number of items that exist within the radius of the distance between a user's point and the closest convergence point is 3.3. This closesness between points indicates that the recommendations made to each user is personalized to a significantly greater extent than they would be if the recommendations were made solely based on popular items.

When the embedding space is partitioned in halves in this manner with a convergence threshold of 30, the quiz converges after a minimum of 7 questions and a maximum of 13 questions. These numbers are not hard limits, and the worst-case time complexity for the algorithm is $O(n)$, in which n is the number of embeddings in our embedding space. However, this outcome is extraordinarily unlikely. The best-case scenario in which all branches have converged can be expressed as $n * 0.5^x < k$. In this case, k represents the threshold for a cluster to converge.

## 5   Perspectives & Future Work

This study presents an initial implementation of procedurally generated style quizzes, and there are, therefore, several aspects of it with significant potential for improvement. Notably, more sophisticated traversal methods within the embedding space and refinements to the representation of the users. A significant challenge in this domain lies in the evaluation of the efficacy of such a style quiz assessment tool, as it necessitates direct user feedback, which is a common dilemma within the field of RSs. These issues could explain the lack of existing academic literature on this topic, despite the prevalence of style quizzes on fashion retail sites, as evaluation presents a substantial barrier to entry that is difficult to overcome without access to a live customer base.

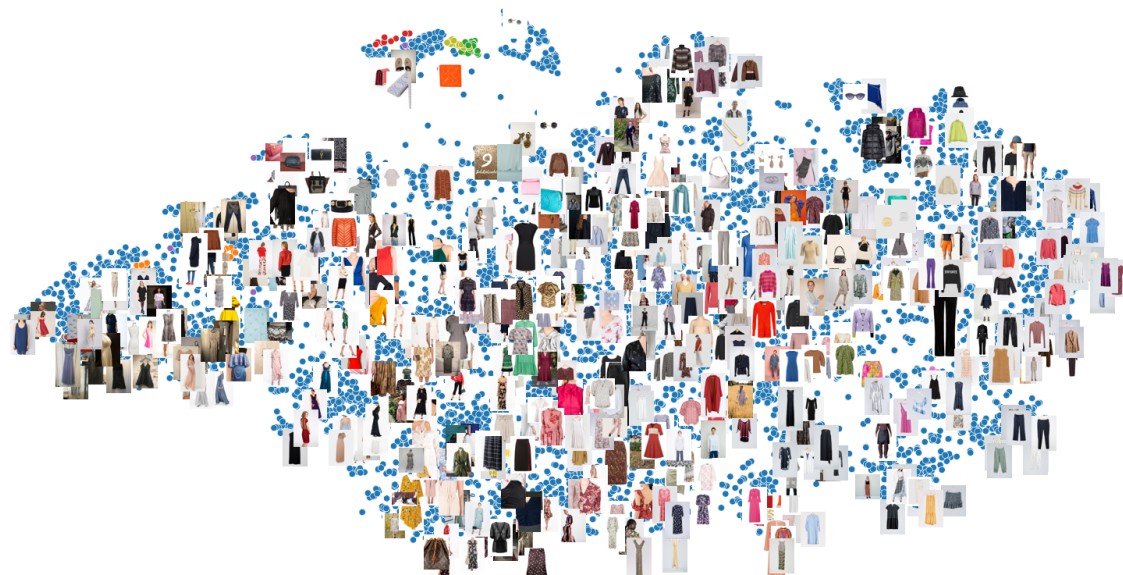

**Figure 3.** A visual representation of a sample of the dataset's images and their corresponding point in the embedding space. This figure complements Figure 2 by providing concrete visual examples of items within the embedding space, allowing for a more intuitive understanding of the embedding space's structure and the types of items found in different regions.

# 6 Conclusion

We have presented Style-Quiz, a novel method for onboarding new users in the context of the user-based extreme cold start problem by generating a style quiz off of embeddings native to a content-based RS. To our knowledge, this article is the first to discuss the construction of style quizzes as a method for alleviating the cold start problem in RSs.

The quiz tasks the user with selecting their preferred style based on a limited set of possible options. Each choice gradually narrows down the searched space until a point in the embedding space is chosen as the user's initial representation. Our results indicate a significant improvement in personalization as compared to the recommendation of popular items. This improvement compared to the recommendation of popular items is particularly important in the context of fashion rental because it alleviates the issue of competition between users for renting the same items.

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
