# OpenReview forum: "Style-Quizzes for Content-Based Fashion Recommendation in Extreme Cold Start Scenarios"
_NLDL.org/2025/Conference — Submitted to NLDL 2025_

### Official Review · Reviewer_uEC4 · 2024-09-17
**NLDL 2025 review for Style-Quizzes for Content-based fashion recommendation in extreme cold start scenarios**

**Confidence:** 3

**Summary:**

This work presents a method called Style-Quiz, in which the authors presents a method for dealing with so-called extreme cold start problems in recommender systems. The cold start problem itself comes from how recommender systems work by using previous user-data to recommend for example products for the user to buy. If we don't know much about the user, then it's quite hard to recommend anything, thereby causing the cold start problem which occurs during on-boarding of the user.

The article present itself as introducing a concept used in industry, and not so much in academia. Some brands such as Nordstorm and Stitch Fix uses these personalized quizzes to onboard new customers, and this paper seeks to introduce and explore this concept into academic circles.

The main method of the paper uses a quiz which, through user input, converges to different convergence points which are presented as cluster centroids for 30 different items.
This convergence point is then used as the customers "initial state", if that would be a correct way of stating that, which is much more personalized versus using the global most popular items as an initial point.

**Strengths:**

The introduction set up an interesting story and I would consider this section solid.
The problem and key terms is clearly presented and explained in such a way that those not familiar with the recommender system subfield can still understand what is being explained and understand the novelty of the articles findings.


The methods section presents clearly what dataset has been used and where to find the code for this work (I assume, the link is anonymized).

**Weaknesses:**

Related Works

This section comes off as a bit short and lacking. I'm not going to hold this to much against the author as there doesn't seem to be that many works addressing the cold start problem. Maybe some other works into recommender systems or metrics used could be placed here, but I'm not familiar enough with the field to know exactly what could be included here. I'm mainly placing this comment here as there isn't a dedicated "comments" tab for this reviewing software.



Method

The methods section lacks explanations into how the authors will approach the question of by what metric do we determine the "goodness" of the method, and what can we compare it to.

One thing missing in this section is how to measure any sort of "goodness" or improvement over other methods. Other papers might use accuracy, AUC, R^2 values, or any other sense of measurements to argue for their method, but especially for this work it's not obvious how a good method should look like and how to measure this. This hurts the article in my mind as when we get to the results section we'll see some assumptions on a models goodness come forward which at that point has not been discussed in depth.

Paragraph 2 of section 3.1 could use a small clarification on "biasing towards similarity between outfit categories". It’s not clear why we’d want to do this. My guess would be that recommender systems tend to recommend similar objects while users want diversity as shown using the Simpson's Diversity Index. It would be nice to have this specified.


In Section 3.2 the work presents a scheme to segment the data into smaller and smaller sub-clusters of similar items until each cluster has 30 samples. It's mentioned that this number is arbitrary, which makes sense, but here it would be nice if the authors justified this specific number. Is it somehow derived from the dataset, or is this somewhat common practice? If this number is not derived from the dataset in any way, could it be?

It's also not exactly clear how the quiz would be set up. Does this have an effect, or can it be set up in several different ways? This should be explained.



Results

The results section suffers from a lack of buildup from the methods section. The only real metric considered in this section is mean distance of points from points in the dataset to either the closest convergence points or the global most popular cluster. The authors show that their method reduces the mean distance to the closest cluster representative.
Intuitively we can understand that this would lead to a more personalized initial start, but this way of assessing "goodness" has not been explained, and is not explored further. See for example source [13], where they address extreme cold start scenarios and take a section to explain their metrics and what it will show.

The cluster size is touched upon, but not further explored.

The TSNE plots needs a bit more explaining and exploration. It's not clear why only a few clusters in Figure 2 are colored, and it's hard to know if Figure 3 is there to make a point or not.

**Justification:**

This work does present an interesting view into a problem which has not seen much exploration to what I can find looking for similar work.
The problem setting is interesting, and the proposed solution sounds like it makes sense.

The article does however fall short in its exploration and explanation of the method it presents. Too many questions are left up in the air by the end of the paper. How is the quiz structured, why are the metric presented considered a good metric for this task, why are there no comparisons to other methods dealing with the similar problem, why do we not see a more in depth exploration into this method, etc?

The introductions sets the stage for this paper to tell a great story, but the following sections lacks the substance, exploration and rigor to make this article an academic piece.

---

> ### Author Rebuttal · Authors · 2024-10-17
>
> Thank you for review, we sincerely appreciate the thorough feedback and will keep your points in mind for later iterations on the article.

---

### Official Review · Reviewer_d8jo · 2024-09-17

**Confidence:** 4

**Summary:**

The authors present a sequential clustering-based approach to identify items of interest of a user to remedy the cold-start problem. The idea is to gradually reduce the candidate set of items until only a few remain by a series of questions which are used to dive into subsets of the item pools.

**Strengths:**

Interesting and challenging problem that has many degrees of freedom.

**Weaknesses:**

There is unfortunately no technical contribution and also no quantitative evaluation. It does not become obvious whether the approach works out, there are no baselines to compare with etc. Many design choices are not motivated well, e.g., how many selections are presented to the user, why is there only one choice per question, etc. Model selection seems necessary.

**Final Rebuttal Confidence:**

5

**Final Rebuttal Justification:**

To me this is a clear reject. More than one reviewer commented on the lack of experimental evidence and the authors didn't actually propose to do so but call it future work. For a workshop, this would certainly be OK but for a conference we need more evidence to support the idea. From the rebuttal, I would also assume that they understood why the paper is going to be rejected and just replied for politeness.

**Justification:**

This may be an interesting poster at a workshop but it is not matured yet enough for a conference.

---

> ### Author Rebuttal · Authors · 2024-10-17
>
> Thank you for your review. As we mentioned in the paper, validating the method's performance is likely one of its key technical challenges. Despite this, we believe that this initial publication could serve as a valuable starting point for later research on this topic.

---

### Official Review · Reviewer_Syhz · 2024-10-07
**Modification of experiments and methods**

**Confidence:** 3

**Summary:**

The paper introduces Style-Quiz, a method for solving the user-based cold start problem in content-based recommender systems. Meanwhile, this method can obtain user preference information in the absence of user history.

**Strengths:**

1. The paper is well thought out.
2. The introduction to the concept is complete.

**Weaknesses:**

1. Style testing has been used in many scenarios and is well developed, but the paper's approach is too simplistic.
2. The comparison experiments only compare the set of experiments that recommend the most popular items, which is not enough to show that this method is good enough.
3. Experiments should use multiple datasets, using only one is not generalizable.

**Justification:**

1.The method of use is simple.
2.Comparing experiments and datasets there is just one.

---

> ### Author Rebuttal · Authors · 2024-10-17
>
> Thank you for your review.

---

### Official Review · Reviewer_KEpS · 2024-10-10

**Confidence:** 5

**Summary:**

In this submission, the authors design a style-quiz, helping the users of recommender systems identify their own fashion style preferences in cold-start scenarios. A hierarchical tree is generated for the style embedding of different items, and the users answer the questions in the quiz from the root of the tree to the leaves, and the systems get their preferences from a coarse to a fine level.

**Strengths:**

1. The idea is simple but reasonable.

2. The writing of this paper is clear.

**Weaknesses:**

My main concern is the lack of experiments:

1. Cold-start recommendation is a classic and challenging problem for recommendation system, and many methods have been proposed to solve the problem. However, this work neither provides any experimental result in the main paper nor analyzes the differences between the proposed method and existing ones. Without solid experiments and analysis, the rationality and advantages of the proposed method are not convincing.

2. In practice, the efficiency of the quiz mechanism is important and should be analyzed in detail. For example, how many selections should a user make? How long does it take? Is there any trade-off between the number of selections and the recommendation performance? Without such analysis, this submission is not solid enough.

**Justification:**

Although providing anonymous code, this submission did not show any comparisons or analytic experiments in the main paper, making the rationality of the proposed method questionable.

---

> ### Author Rebuttal · Authors · 2024-10-17
>
> Thank you for the review, there are absolutely many other aspects that should be evaluated for this approach. The primary intention of this paper has been to point out a niche method employed by the industry, but which has seen no representation within the machine learning literature. Therefore we believe that while our evaluation methods are lacking, it provides a valuable starting point for further research on this topic.

---

### Meta-Review · Area_Chair_vnZo · 2024-10-31

**Recommendation:** Reject
**Confidence:** 4

**Metareview:**

This paper introduces a StyleQuiz to tackle the cold start problem in recommender systems. The reviewers agree that the paper is well written, and the problem addressed is interesting and relevant. However, they express concerns about the insufficient evaluation to fully support the effectiveness of the proposed approach and the need for a more detailed explanation of the method and its design.

Therefore, the current version of the paper is not yet ready for publication. We encourage the authors to address the reviewers’ concerns to enhance the manuscript’s quality.

**Suggested Changes To The Recommendation:**

3: I agree that the recommendation could be moved up

---

### Decision · Program_Chairs · 2024-11-06

Reject